# CO$_2$ Emissions from Soils under Different Tillage Practices and Weather Conditions

**Gabriela Mühlbachová \*, Pavel Růžek, Helena Kusá and Radek Vavera**

Crop Research Institute, Drnovská 507/73, Prague 6-Ruzyně, CZ 161 06 Prague, Czech Republic; ruzek@vurv.cz (P.R.); kusa@vurv.cz (H.K.); vavera@vurv.cz (R.V.)

**\*** Correspondence: muhlbachova@vurv.cz; Tel.: +420-702-087-748

**Abstract:** CO$_2$ emissions are one of the greenhouse gases that significantly contribute to climate change. The use of reduced soil tillage practices could contribute to the mitigation of CO$_2$ emissions from soils under ongoing climate change conditions. The use of reduced and no-tillage practices in the summer period, the most critical period for CO$_2$ and for water loss from soils, would contribute to the mitigation of CO$_2$ emissions that is required by the European Union. The aim of this research was to contribute to the specification of CO$_2$ emission factors, following different soil tillage practices in the summer period under variations in weather. Gentler tillage practices were defined in terms of reducing CO$_2$ emissions from the soil. This research was carried out as a long-term field experiment. The effects of soil tillage practices on CO$_2$ emissions were studied over a six-year period as a long-term field experiment and concerned the use of different soil tillage practices for over 20 years (established in 1995), with these including conventional tillage (CT; plowing to 20–22 cm), reduced tillage (RT; chiseling to 10 cm), and no-tillage (NT; without tillage). The crop rotation was winter wheat–winter oilseed rape–winter wheat–pea. CO$_2$ emissions were measured at least 7–10 times during the summer–autumn period in the years 2017–2022 after agrotechnical operations following the winter wheat harvest. Soil moisture was determined in all the treatments. Weather conditions were measured by means of the meteorological station of the Crop Research Institute. The CO$_2$ emissions were the highest in the summer period under CT in comparison with RT and NT. Reduced tillage and no-tillage practices, with mulch on the surface of the soil, decreased CO$_2$ emissions by a 6-year average of 45% and 51%, respectively. The mean CO$_2$ emissions were 6.1, 3.1, and 2.9 μmol CO$_2$ m$^{-2}$ s$^{-1}$ for CT, RT, and NT. The highest CO$_2$ emissions and the largest differences among different tillage practices were measured in 2019, with high temperatures and repeated rainfall. CO$_2$ emissions under CT reached 22 μmol CO$_2$ m$^{-2}$ s$^{-1}$, which was 7.5 and 5.8 times higher than under RT and NT, respectively. Current weather conditions, mainly temperature and precipitation, played an important role in CO$_2$ emissions. The hot and dry weather in 2018 decreased overall CO$_2$ emissions, with CO$_2$ emissions, even under conventional tillage, reaching only 2.5 μmol CO$_2$ m$^{-2}$ s$^{-1}$ on average. As a result of climate change, the temperatures also gradually increased in the later stages of the year, with more summer days being expected during autumn and higher CO$_2$ emissions from soils being expected as a result.

**Keywords:** plowing; reduced tillage; no-tillage; moisture; straw residues; weather; CO$_2$-fluxes

## 1. Introduction

Ongoing climate change brings about increasing temperatures, more frequent periods of drought, and excess precipitation, all of which represent a great challenge for new approaches and technologies in agriculture. The Intergovernmental Panel on Climate Change (IPCC) predicts an increase in global temperatures of at least 1.5 °C by the end of the 21st century [1]. A significant increase in air temperatures in past decades has been extensively documented in the Czech Republic, as well as in other European countries. Since the year 2000, six major droughts within Central Europe have been reported, including

the spring drought in 2000, the well-known 2003 drought, two more regionally constrained events during 2006–2007 and 2011–2012, and the 2015–2019 drought episode, which can be classified as a 500-year drought [2–4]. In fact, Central Europe has experienced a sequence of unprecedented summer droughts since 2015, which has had considerable effects on the functioning and productivity of natural and agricultural systems [5]. In addition, days with extremely high temperatures are becoming more frequent as a consequence [6].

Some approaches in agriculture, including intensive tillage, together with higher temperatures and other weather events, contribute to the loss of organic matter in soil, increase $CO_2$ emissions to the atmosphere, and, as a consequence, result in a loss of soil fertility, the disruption of soil aggregates, and soil losses due to increased erosion [7,8]. Soil $CO_2$ emissions increase with the increase in soil temperature as a result of seasonal soil temperature variation [9], more frequent drought periods, and excess precipitation. This temperature increase leads to higher mineralization and losses in soil organic matter under conventional tillage, in contrast to conservative tillage systems [10]. In fact, the mineralization of soil organic matter depends on temperature, precipitation, the chemical composition of plant residues, the structure and composition of microbial communities, and the C:N ratio [11]. With the recent weather changes, the stabilization and rebuilding of C stocks in agricultural soils requires the adoption of less intensive management practices, which can help mitigate $CO_2$ emissions and maintain acceptable crop yields [12,13]. Conservative tillage practices should not only decrease $CO_2$ emissions but also improve soil water and temperature regimes, together with their interactions, in addition to the availability of substrates [14,15].

For all of these reasons, more friendly soil tillage practices with reduced or no-tillage systems are increasingly used as substitutes for traditional plowing. No-tillage systems, including mulching materials or organic fertilizers, can help improve the soil organic carbon (SOC) and ultimately crop grain yields as well [9].

Sustainable land management systems that could be used in the coming years are associated with a better retention of water, carbon, and nutrients in the soil, which can be achieved through the use of new cultivation practices. $CO_2$ emissions from soils depend on many factors, including the soil tillage practice used, current weather, soil moisture, temperature, and seasonal variation [9,15]. The amount of precipitation and, mainly, drought periods, affects the decomposition of post-harvest residues [16] and, consequently, $CO_2$ emissions from soils. In addition, the management of crop residues is important for the maintenance of soil organic matter and, depending on the tillage practice used, also contributes to $CO_2$ emissions from soils. Tillage practices are also essential for the soil water regime and the contact area of crop residues and soil [17]. The long-term incorporation of crop residues in soils also increases the content of small macroaggregates (>250 μm) in the surface layer of soil, leading to the accumulation of high carbon and nitrogen concentrations, which, in turn, serve as a substrate for microbial growth and subsequent $CO_2$ emissions [18,19]. The straw remaining on soil surfaces can substantially decrease the $CO_2$ emissions from soils in comparison with conventional tillage [17,20,21]. The crop residues remaining on the soil surface also affect various soil systems due to improved water infiltration, increased moisture retention, a reduction in soil erosion, a decrease in soil temperature, and better weed control [22,23].

The Commission Decision in 2010 [24] introduced emission coefficients for different tillage practices. This decision does not precisely define the periods for tillage, but it contains $CO_2$ emission factors for different tillage practices. Obviously, there are differences between plowing carried out in the summer and that carried out in the later autumn period. The mitigation of $CO_2$ emissions from soils and the effort to accumulate soil carbon via reduced or no-tillage practices is in conformity with the Commission Implementing Regulation (EU) 2022/996 [25]. The summer period is the most vulnerable to $CO_2$ emissions from soils under tillage. This research evaluated $CO_2$ emissions in a six-year summer period under weather conditions of a given year and under different soil tillage practices. There are still not enough data on $CO_2$ emissions regarding the soil tillage in the critical summer period. During the summer, temperatures are high, and deeper soil cultivation, necessary



for growing crops needing early summer sowing (i.e., oilseed rape), results in high $CO_2$ emissions and water losses from the soil. These facts are not sufficiently taken into account by the Commission Decision [24].

The aim of the research was to define the differences in $CO_2$ emissions and water losses from soil under different tillage and weather variations during the critical summer period for agrotechnical operations on soils in order to decrease the carbon footprint of crops sown in summer. The obtained results will serve as useful data at the national ministerial and European levels to specify emission factors for different soil tillage practices in the summer.

## 2. Materials and Methods

### 2.1. Site Description

A long-term field trial was established on a permanent arable field in 1995 at the Crop Research Institute (CRI) of Prague (Czech Republic: 50°05′ N; 14°17′ E). The climatic region is warm and moderately dry. The altitude is 360 m above sea level, the mean annual temperature during the studied period (2017–2022) was 9.8 °C (ranging from 8.3 °C to 11.1 °C), and the mean annual precipitation was 516 mm (ranging from 345 mm to 731 mm; Crop Research Institute Prague-Ruzyně Meteorological Station). The soil type is classified as illimerized luvisol [26]. The parent material is loess mixed with highly weathered chalk [27]. The soil texture is silty clay loam, with pH (0.01 M $CaCl_2$) 7.0, pH ($H_2O$) 7.8, and SOC 1.3%; the texture is sand 15.8%, silt 54.9%, and clay 29.3%. The average nutrient content extracted by the Mehlich 3 method was: P—61 mg kg$^{-1}$; K—176 mg kg$^{-1}$; Ca—3650 mg kg$^{-1}$; Mg—137 mg kg$^{-1}$; and CEC—191 mmol kg$^{-1}$.

### 2.2. Field Trial

The long-term experiment started in 1995 in a field, which until then had been conventionally tilled. Two main blocks (95 m × 95 m) were established. After that, the three tillage practices (CT = moldboard plowing down to 20–22 cm; RT = chisel plowing of the surface soil layer to a depth of 10 cm; and NT = without tillage) were applied in each field block in strips with a width of 12 m and a length of 95 m. A Kvernland ES 95 reversible plow (Soest, Germany) was used for moldboard plowing (CT). A Lemken Smaragd 7 two-arm sweep cultivator (Alpen, Germany) was used for chiseling (RT). The following crop rotation was alternately applied in each of the blocks: winter wheat—pea—winter wheat—oilseed rape. All post-harvest crop residues were always incorporated into the soil according to the used tillage practice—completely under CT, partly under RT, and remaining on the surface under NT as the mulch covering the soil surface. Phosphorus and potassium fertilizers were applied in autumn on the whole experimental field at the same dosage. The fertilizer doses were as follows: phosphorus as diammonium phosphate—52 kg $P_2O_5$ ha$^{-1}$, and potassium as potassium chloride—40 kg $K_2O$ ha$^{-1}$. Nitrogen fertilizers were applied only during the winter wheat growth as calcareous ammonium nitrate at a total dose of 120–140 kg N ha$^{-1}$, according to the mineral nitrogen content in the soil and the calculated yields. Pesticides were applied during the winter wheat growth uniformly over the entire experimental field. The last application was that of fungicide (bifaxen, prothiconazole, spiroxamine) carried out on the BBCH 37 at least 2–2.5 months before harvest.

### 2.3. CO₂ Measurements

The $CO_2$ emissions from soils were measured in the years 2017–2022 after the wheat harvest in a given field block with the performed soil tillage (CT, RT, or NT). The dates of soil tillage were 23 August 2017, 11 August 2018, 17 August 2019, 16 August 2020, 23 August 2021, and 17 August 2022. $CO_2$ measurements started one or two days after tillage and then at approximately one-week intervals according to the weather conditions from the second half of August to the end of October. The measuring sites were chosen during the winter wheat vegetation when the evenness of the wheat growth was evaluated in order to avoid damaged places. The rings were placed according to the movement of machinery to avoid the paths overlapping and wheel tracks. The same measuring locations

were used in the field during the whole measuring period of each given year. Original LI-COR rings were used. The $CO_2$ emissions were measured by means of a LI-COR 8100 Automated Soil $CO_2$ Flux system equipped with an 8150 Multiplexer and three long-term chambers (LI-8100-104, Lincoln, NE, USA) (diameter 20 cm) operating in an open–closed system. Four independent $CO_2$ measurements were carried out for each tillage technology during each measurement day in the regime of a 2 min closed chamber, with the start of reading after 25 s, a 45 s purchase time after opening, and the consecutive changing of treatments. The gravimetric soil moisture was measured in soils sampled in 0–10 cm depth by drying 20 g of the soil at 105 °C for 5 h to a stable weight.

### 2.4. Temperature and Precipitation

The daily air temperatures and precipitation were measured by means of the meteorological station of the Crop Research Institute. The measurements are provided in 15 min intervals, and the daily temperature averages and precipitation sum are calculated.

### 2.5. Statistical Analysis

Statistical calculations were performed using the Statistica 14.0 Software (TIBCO, Palo Alto, CA, USA). The results were expressed as the mean values for each treatment and measurement. One-way ANOVA was used to evaluate the effects of treatments. The same letters in the figures indicate statistically identical values according to Tukey's test ($p < 0.05$). The statistical evaluation was performed for each sampling day separately.

## 3. Results

### 3.1. Weather Conditions

The weather conditions differed in each studied year (Figures 1 and 2). The periods with hot dry weather were noted mainly in the years 2017–2019. The temperature in the study period in the majority of the studied years exceeded during the summer the average temperature recorded in the years 1995–2014. Zahradníček et al. [6] defined summer days as when the temperature exceeds 25 °C during the day and tropical days as a temperature higher than 30 °C.

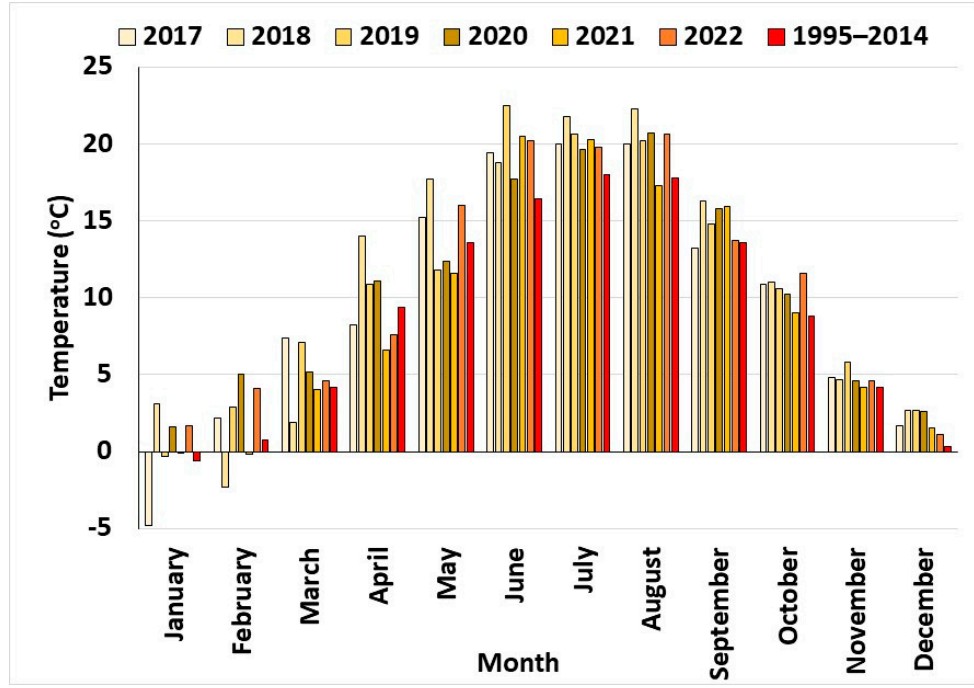

**Figure 1.** Monthly temperature means in the years 2017–2022.

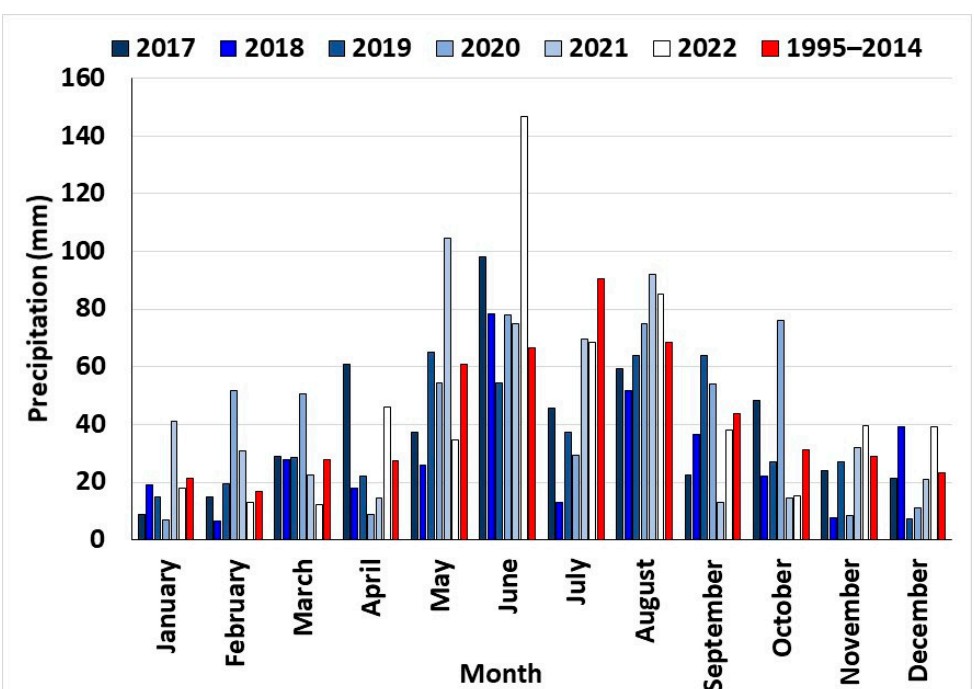

**Figure 2.** Monthly sums of precipitation in the years 2017–2022.

The summer days in the period of 2017–2022 varied. The year 2018 was characterized by high temperatures represented by 59 summer and 30 tropical days, and very low precipitation was noted during the whole vegetation season. On the other hand, the weather in the year 2021 was atypical. The mean monthly temperature in August was about 2 °C lower in comparison with the long-term average, and only 24 summer and 4 tropical days were observed. In the other years, the number of summer days varied between 33 and 40, and 3–14 days were classified as tropical.

Higher temperatures were observed in the period of 2017–2022 in comparison with the 1995–2014 average. Typical summer temperatures were observed in September, and higher temperatures than average were also observed in October.

The distribution of precipitation in the studied years differed among the years (Figure 2). We noted periods of drought followed by high precipitation. A long drought in the year 2018 affected the microbial activities and soil mineralization processes in the soils. In addition, irregular distribution of precipitation was noted during a particular month when it occurred. For instance, the precipitation in April was low in four out of the six studied years. Sufficient precipitation in April is important for growing crops. High precipitation in May 2021 followed a lack of rainfall in March and April 2021. Irregularly high precipitation was noted also in June 2022 when the sum of precipitation between 1 and 23 June was 39 mm compared to the last week of June, for which it was 108 mm. In all years, the precipitation in July was lower than the long-term average from 1995–2014. The distribution of precipitation in August differed among the years. Lower average precipitation was observed in the years 2017–2019. From 2020, a higher but irregular level of precipitation was noted. For instance, between 1 and 14 August 2022, the sum of precipitation was 22 mm in two rainfalls, and 63 mm was noted between 19 and 27 August.

*3.2. Soil Moisture*

The soil moisture depended on the precipitation one week before measurement (r = 0.276, *p* < 0.001), the year of measurement (r = 0.490, *p* < 0.001), and the soil tillage practice (r = 0.270, *p* < 0.01). The temperature did not play a significant role, as the correlations were only r = 0.111 and 0.090 forthe temperature on the day of measurement and one week before the measurement, respectively. Table 1 shows the mean, maximum, and minimum humidity in

soils under different tillage practices during summer–autumn measurements. The scarcity of precipitation observed in the year 2018 caused a decrease in moisture to under 10%, which was visible under all tillage practices. In the years 2021 and 2022, the soil moisture tended to decrease during the autumn, which can be partly related to the lower precipitation and the air temperatures, which were higher in comparison with the previous long-term period. The soil moisture was mostly the highest under the NT practice, followed by the RT and CT practices (Table 1). In fact, the straw mulch and non-disturbed soil under NT prevented excess water evaporation, which led to the maintenance of higher soil humidity under dry conditions (for example in 2018) and also to lower $CO_2$ emissions from soils. Conventional tillage caused water losses in all the studied years and the lowest soil moisture.

**Table 1.** The mean, maximum, and minimum soil moisture under different soil tillage practices during summer–autumn measurement. The differences represent the standard error.

| | | | Moisture | (%) |
|---|---|---|---|---|
| Year | Tillage | Mmean | Mmax | Mmin |
| 2017 | Conventional till. | 14.5 ± 2.1 | 17.3 ± 0.4 | 10.8 ± 0.1 |
| | Reduced till. | 16.7 ± 2.2 | 20.0 ± 0.3 | 14.0 ± 0.2 |
| | No-till. | 18.3 ± 3.1 | 23.7 ± 0.7 | 14.1 ± 0.3 |
| 2018 | Conventional till. | 12.0 ± 1.1 | 15.5 ± 0.1 | 8.1 ± 0.2 |
| | Reduced till. | 12.5 ± 1.7 | 16.6 ± 0.1 | 7.8 ± 0.4 |
| | No-till. | 14.1 ± 0.7 | 18.5 ± 0.4 | 9.1 ± 0.3 |
| 2019 | Conventional till. | 15.4 ± 2.5 | 18.5 ± 0.2 | 12.2 ± 0.1 |
| | Reduced till. | 16.7 ± 2.7 | 19.9 ± 0.1 | 12.7 ± 0.1 |
| | No-till. | 18.1 ± 1.7 | 19.9 ± 0.2 | 15.4 ± 0.3 |
| 2020 | Conventional till. | 14.9 ± 2.5 | 18.2 ± 0.1 | 13.0 ± 0.1 |
| | Reduced till. | 15.2 ± 2.5 | 19.6 ± 0.2 | 12.2 ± 0.0 |
| | No-till. | 16.6 ± 3.0 | 20.6 ± 0.1 | 12.5 ± 0.6 |
| 2021 | Conventional till. | 14.0 ± 1.7 | 15.9 ± 0.2 | 11.2 ± 0.2 |
| | Reduced till. | 15.9 ± 2.0 | 18.2 ± 0.2 | 12.4 ± 0.2 |
| | No-till. | 17.3 ± 2.0 | 19.7 ± 0.6 | 14.2 ± 0.6 |
| 2022 | Conventional till. | 17.1 ± 1.4 | 18.6 ± 0.4 | 15.0 ± 0.1 |
| | Reduced till. | 18.0 ± 1.7 | 19.9 ± 0.6 | 15.8 ± 0.2 |
| | No-till. | 18.9 ± 1.0 | 20.0 ± 0.4 | 17.0 ± 0.4 |

*3.3. $CO_2$ Emissions*

The $CO_2$ emissions depended on the intensity of the tillage practice and the current weather conditions (Table 2). Significant correlations were found between the $CO_2$ emissions and tillage, the temperature on the day of the measurement, the temperature the week before the measurement, and the precipitation the week before the measurement (for all: $p < 0.001$).

**Table 2.** Correlations among $CO_2$ emissions, soil tillage, moisture, and weather conditions.

| | $CO_2$ Emissions | Moisture | Temperature—Day of Measurement | Temperature—Week before Measurement | Precipitation—Week before Measurement |
|---|---|---|---|---|---|
| Tillage | −0.340 *** | 0.270 *** | −0.009 | −0.005 | 0.006 |
| $CO_2$ emissions | - | 0.078 | 0.397 *** | 0.416 *** | 0.291 *** |
| Moisture | 0.078 | 1.000 | −0.112 ** | −0.090 * | 0.276 *** |

$* = p < 0.05; ** = p < 0.01; *** = p < 0.001.$

Considering individual soil tillage practices, significant correlations were found between $CO_2$ emissions and soil moisture under CT ($r = 0.200$; $p < 0.01$) and RT ($r = 0.249$; $p < 0.001$). No significant correlation was found between $CO_2$ emissions and moisture under NT, which maintained a higher soil moisture in comparison with CT and RT (Table 3).

**Table 3.** Correlations among $CO_2$ emissions, moisture, and weather conditions under individual tillage practices.

| | $CO_2$ Emissions | Soil Moisture | Temperature Day of Measurement | Temperature Week before Measurement | Precipitation Week before Measurement |
|---|---|---|---|---|---|
| **Conventional till.** | | | | | |
| Year | 0.165 * | 0.524 *** | −0.083 | −0.109 | 0.035 |
| Day | −0.491 *** | 0.009 | −0.766 *** | −0.853 *** | −0.240 *** |
| $CO_2$ emissions | - | 0.200 ** | 0.488 *** | 0.482 *** | 0.386 *** |
| Moisture | | - | −0.132 | −0.111 | 0.314 *** |
| **Reduced till.** | | | | | |
| Year | 0.338 *** | 0.529 *** | −0.084 | −0.112 | 0.036 |
| Day | −0.548 *** | −0.115 | −0.766 *** | −0.853 *** | −0.240 *** |
| $CO_2$ emissions | - | 0.249 *** | 0.456 *** | 0.503 *** | 0.304 *** |
| Soil moisture | | - | −0.103 | −0.048 | 0.343 *** |
| **No-till.** | | | | | |
| Year | 0.183 * | 0.105 | −0.084 | −0.112 | 0.036 |
| Day | −0.609 *** | −0.016 | −0.766 *** | −0.852 *** | −0.241 *** |
| $CO_2$ emissions | - | 0.004 | 0.434 *** | 0.529 *** | 0.309 *** |
| Soil moisture | | - | 0.133 | 0.064 | −0.073 |

* = $p < 0.05$; ** = $p < 0.01$; *** = $p < 0.001$.

The $CO_2$ emissions varied among soil tillage practices and within the studied years (Table 4; Figure 3a–f). Usually, the $CO_2$ emissions were the highest shortly after the soil tillage and decreased during the autumn with decreasing temperatures (Figure 4). The plowing caused mostly higher $CO_2$ emissions compared to less intensive tillage practices. Therefore, the highest $CO_2$ emissions were mostly found under CT rather than under RT and NT. However, the results in the years 2017, 2018, and 2021 also showed that immediately after plowing, $CO_2$ emissions can be temporarily lower than under other tillage practices due to the inversion of deeper layers with lower biological activity and possibly also due to the drying of the upper soil layer. The mean $CO_2$ emissions under RT were lower by 45% and under NT by 51% compared to CT on average during 2017–2022. The $CO_2$ emissions in the year 2017 were the highest under CT with a maximum of 10.7 μmol $CO_2$ m$^{-2}$ s$^{-1}$ during the second measurement (the eighth day after soil tillage) (Figure 3a). The mean decrease in $CO_2$ emissions in 2017 was 58.2% for RT and 33.8% for NT. The lowest difference was found in the hot and dry year of 2018 when the mean $CO_2$ emissions were the lowest of the whole studied period and decreased only by 5.6% under RT and 17.0% under NT compared to CT. The second lowest decrease in $CO_2$ emissions was noted in the year 2020 (12.1% for RT and 35.4% for NT). The lower difference among $CO_2$ emissions under tillage practices could possibly have been caused by the lower temperatures and precipitation at the end of August and beginning of September and by high precipitation after 25 September 2020. On the other hand, the greatest mean decreases in $CO_2$ emissions were noted in the years 2019, 2021, and 2022 (70.8%, 61.4%, and 61.5% for RT and 64.3%., 79.3%, and 74.3% for NT compared to CT).

**Table 4.** Mean, maximum, and minimum $CO_2$ emissions from soils under different types of soil tillage. The differences represent the standard error.

| | | $CO_2$ Emissions (μmol $CO_2$ m$^{-2}$ s$^{-1}$) | | |
|---|---|---|---|---|
| Year | Tillage | Mean | Max | Min |
| 2017 | Conventional till. | 4.63 ± 2.99 | 10.72 ± 4.52 | 1.93 ± 0.19 |
| | Reduced till. | 1.82 ± 0.78 | 3.25 ± 0.91 | 0.89 ± 0.23 |
| | No-till. | 2.86 ± 1.42 | 5.57 ± 3.98 | 1.77 ± 0.34 |
| 2018 | Conventional till. | 2.49 ± 1.10 | 4.42 ± 1.18 | 0.90 ± 0.24 |
| | Reduced till. | 2.39 ± 1.67 | 6.33 ± 0.74 | 0.76 ± 0.28 |
| | No-till. | 1.96 ± 0.72 | 3.31 ± 1.26 | 1.12 ± 0.53 |

**Table 4.** *Cont.*

| Year | Tillage | CO$_2$ Emissions (μmol CO$_2$ m$^{-2}$ s$^{-1}$) | | |
| | | Mean | Max | Min |
|---|---|---|---|---|
| 2019 | Conventional till. | 10.01 ± 8.12 | 22.02 ± 5.55 | 2.07 ± 1.21 |
| | Reduced till. | 2.51 ± 0.96 | 3.64 ± 0.83 | 1.22 ± 0.27 |
| | No-till. | 3.35 ± 1.84 | 7.03 ± 0.48 | 1.36 ± 0.20 |
| 2020 | Conventional till. | 6.38 ± 4.37 | 14.84 ± 3.30 | 1.97 ± 0.61 |
| | Reduced till. | 4.99 ± 3.95 | 14.09 ± 4.21 | 1.78 ± 0.40 |
| | No-till. | 3.44 ± 1.81 | 6.60 ± 1.71 | 1.16 ± 0.27 |
| 2021 | Conventional till. | 5.03 ± 3.42 | 10.92 ± 1.39 | 1.64 ± 0.20 |
| | Reduced till. | 3.56 ± 1.75 | 6.01 ± 1.71 | 1.10 ± 0.19 |
| | No-till. | 2.35 ± 1.36 | 4.97 ± 1.77 | 0.88 ± 0.48 |
| 2022 | Conventional till. | 8.23 ± 4.83 | 18.50 ± 5.75 | 1.60 ± 0.61 |
| | Reduced till. | 4.83 ± 1.93 | 8.52 ± 2.12 | 2.34 ± 1.15 |
| | No-till. | 3.24 ± 1.25 | 5.52 ± 2.75 | 1.44 ± 0.35 |

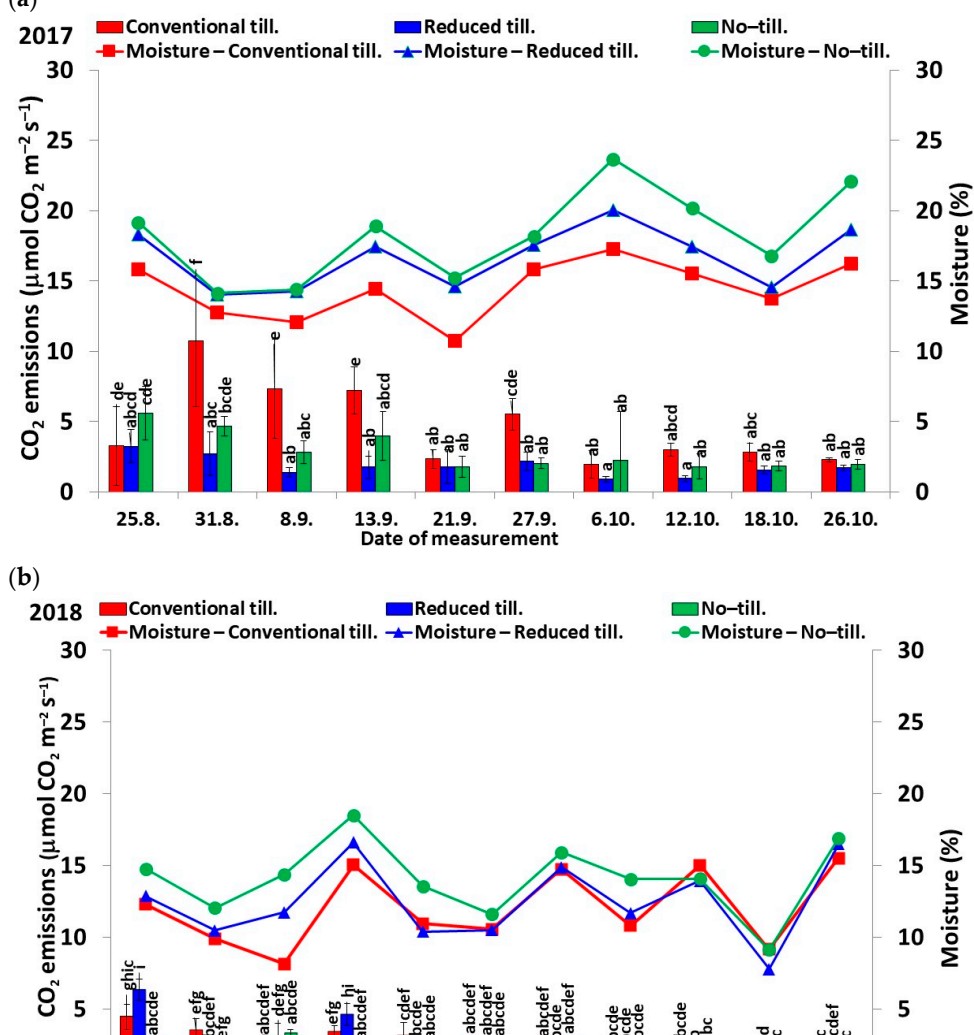

**Figure 3.** *Cont.*

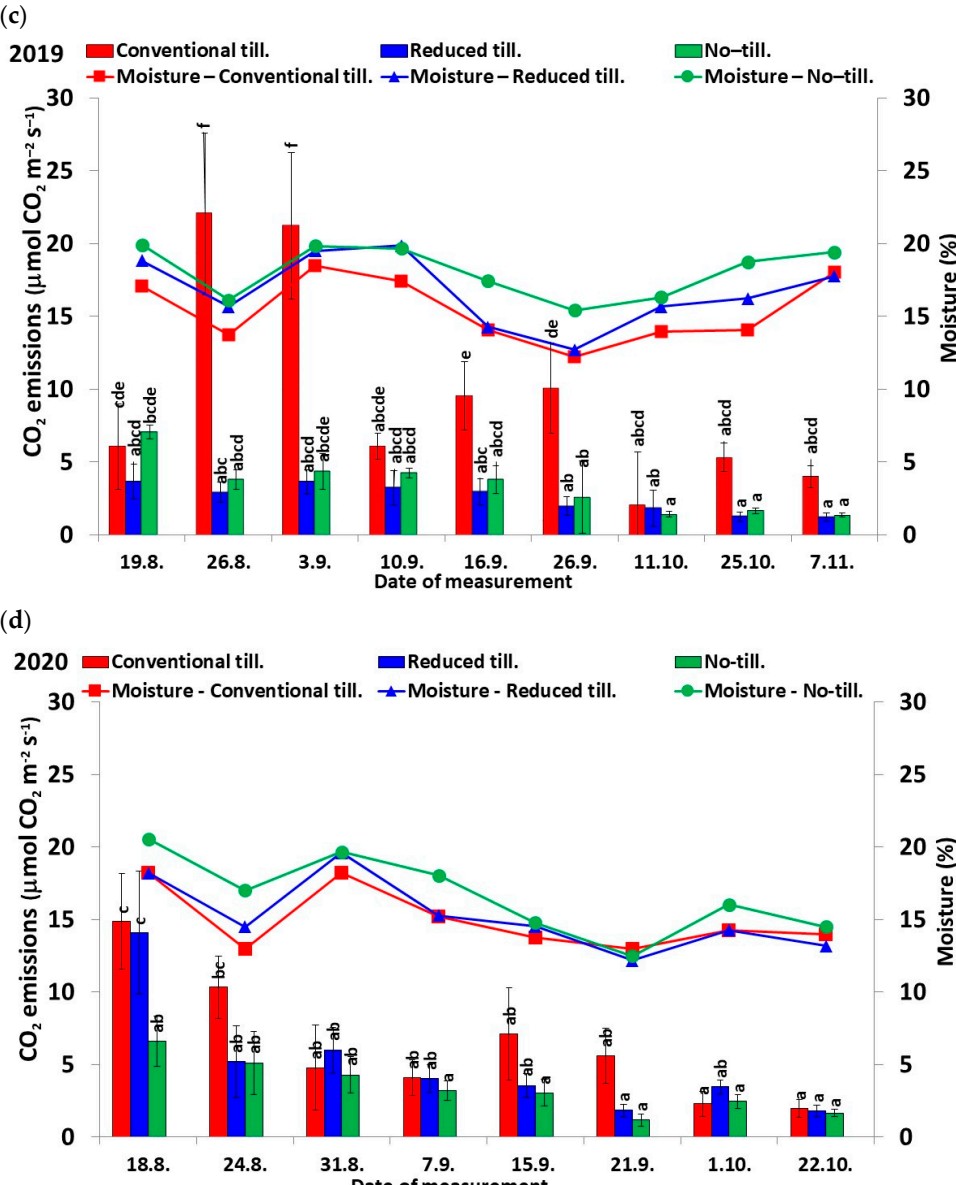

**Figure 3.** *Cont.*

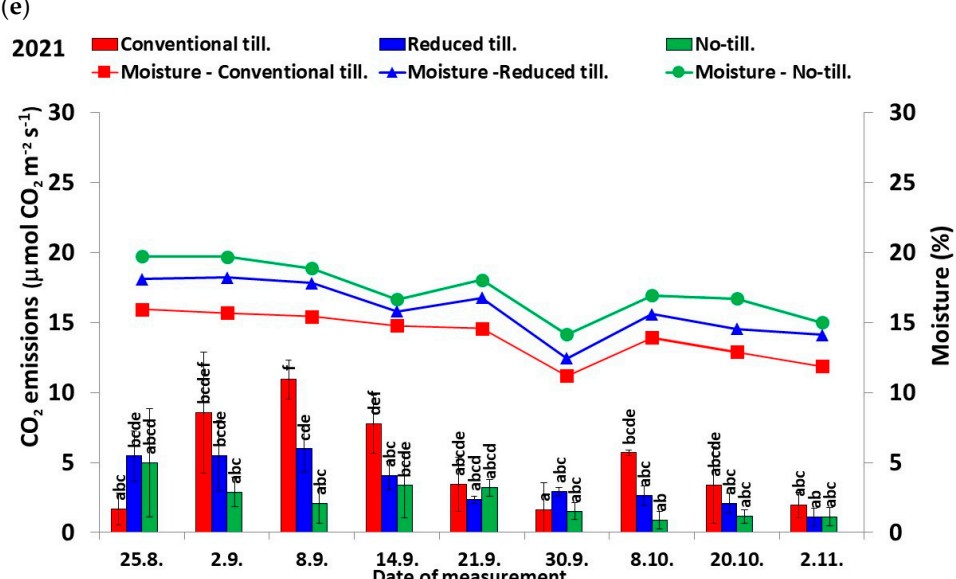

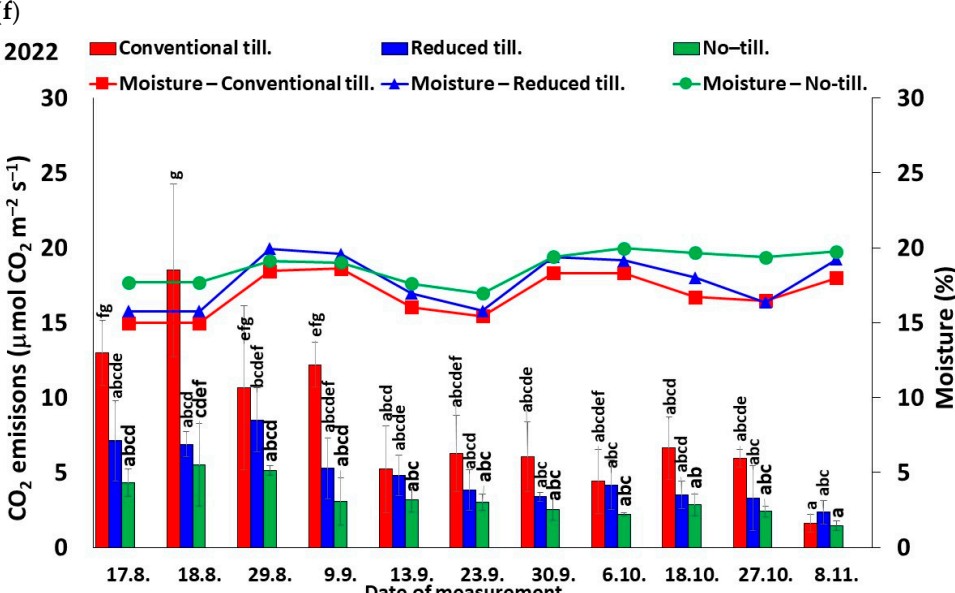

**Figure 3.** $CO_2$ emissions under different soil tillage practices in the years 2017–2022: (**a**) year 2017, (**b**) year 2018, (**c**) year 2019, (**d**) year 2020, (**e**) year 2021, (**f**) year 2022. The bars represent the standard error. Different letters in figures represent the significant differences according to ANOVA.

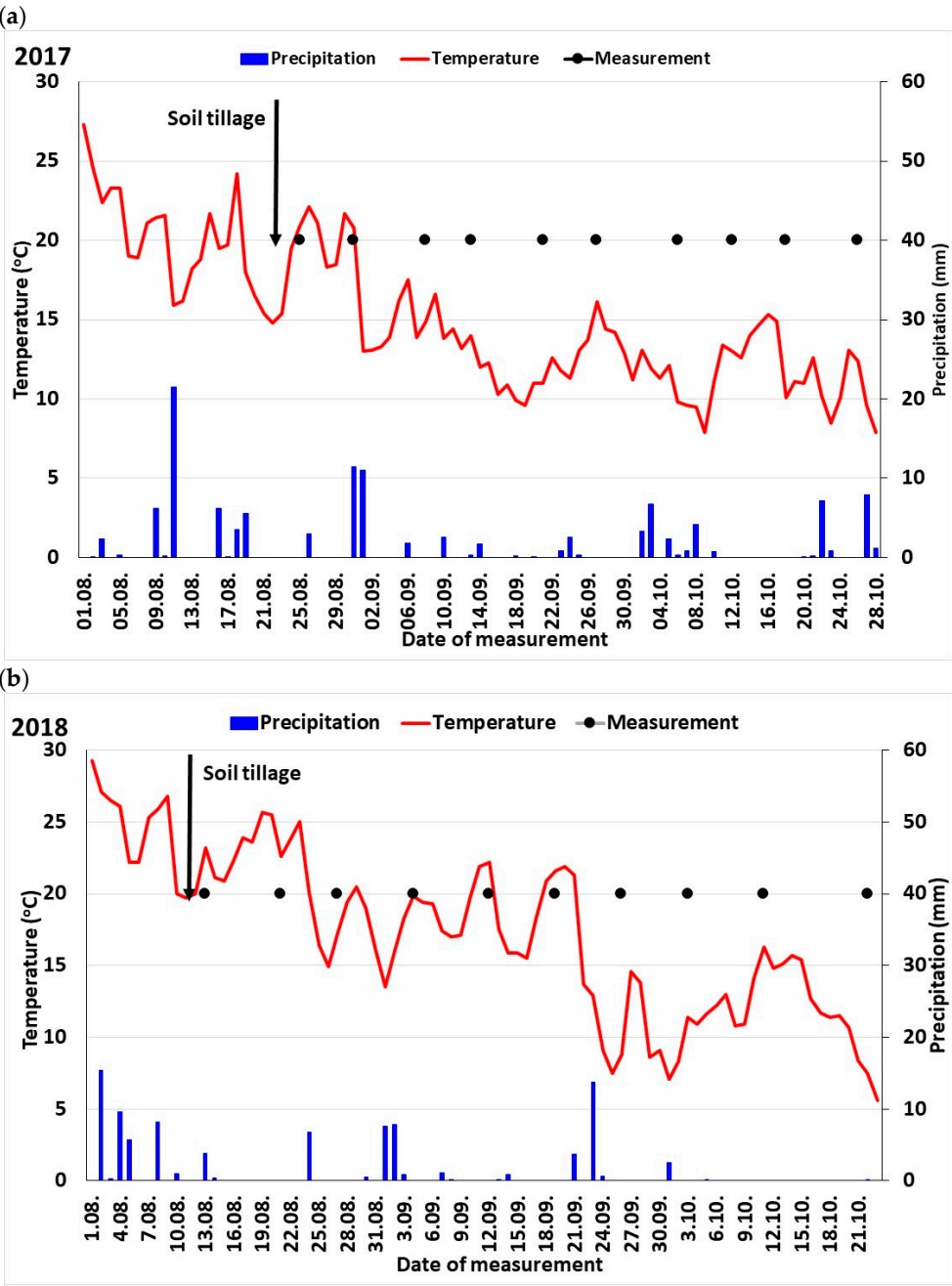

**Figure 4.** *Cont.*

(**c**)

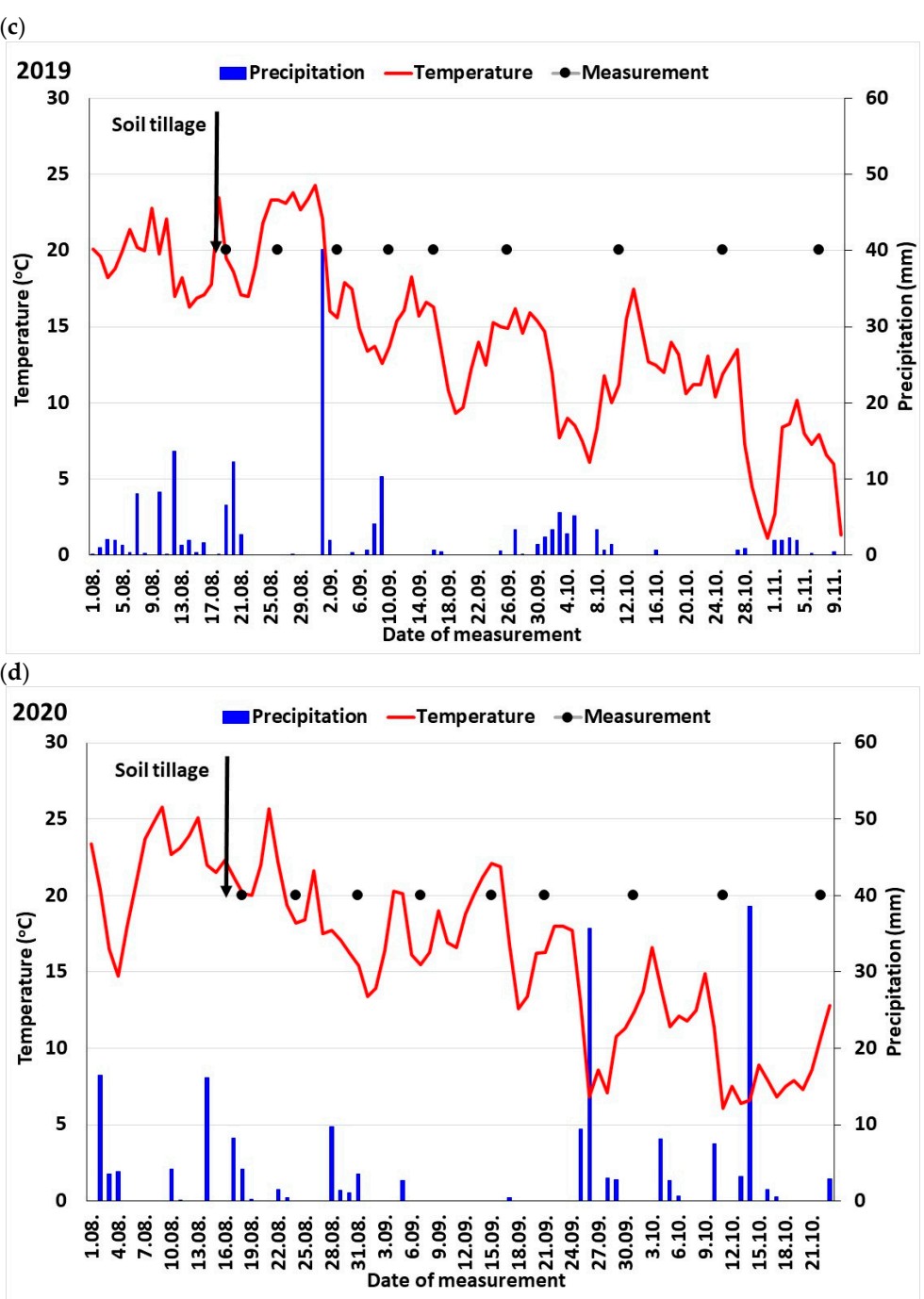

**Figure 4.** *Cont.*

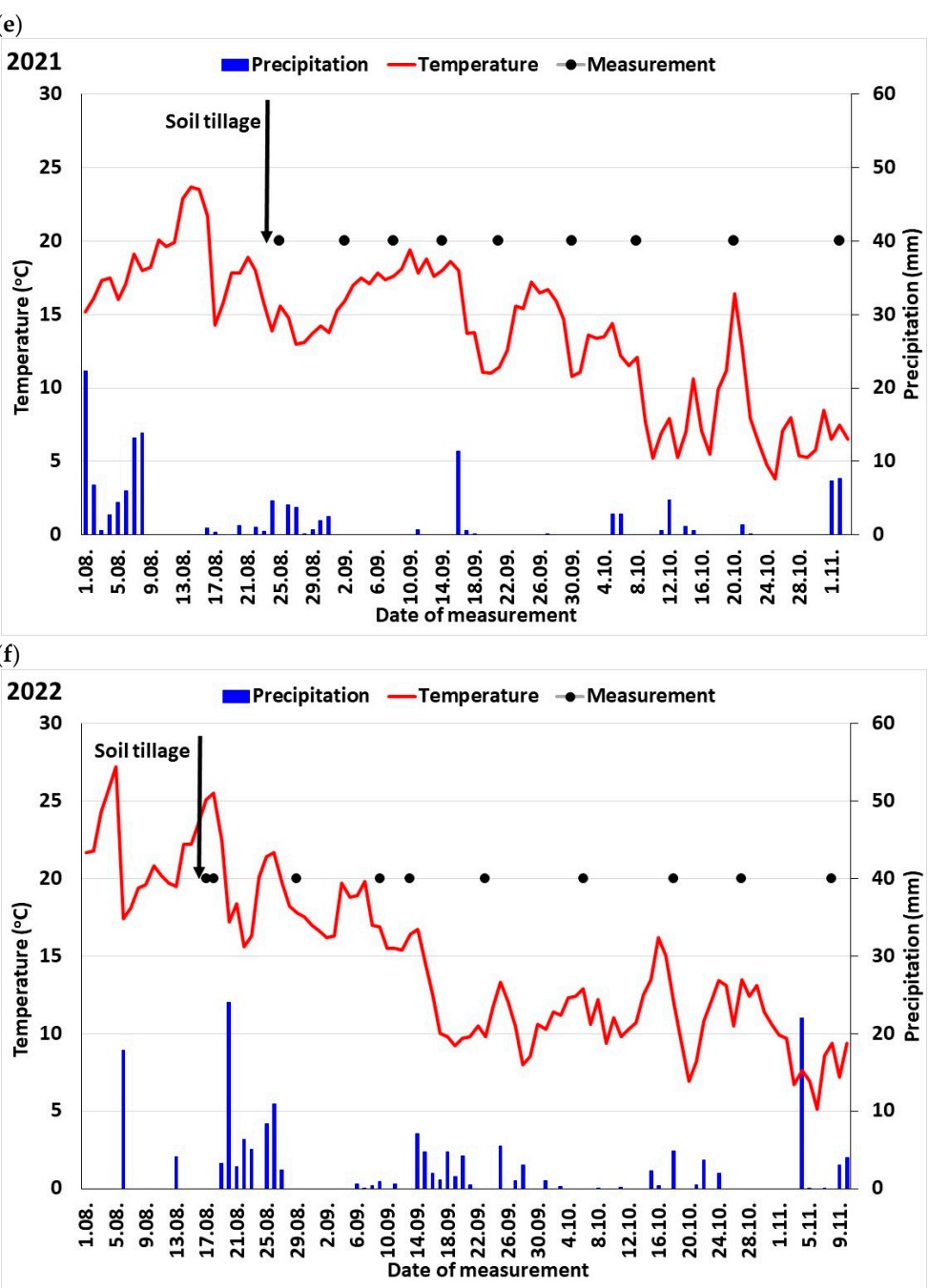

**Figure 4.** Daily temperature means and sums of precipitation during the measurement period and during the summer–autumn period in the years 2017–2022: (**a**) year 2017, (**b**) year 2018, (**c**) year 2019, (**d**) year 2020, (**e**) year 2021, (**f**) year 2022.

Higher soil moisture and warm weather increased $CO_2$ emissions under CT at the end of August and beginning of September 2019. High precipitation (40.1 mm) was noted on 1 September 2019, which in combination with the warm weather probably affected the $CO_2$ measurements at the beginning of September 2019. The $CO_2$ emissions under RT and NT were significantly lower in comparison with CT. On 26 August, the highest value for $CO_2$ emissions over the whole six-year study period was measured. The $CO_2$ emissions under CT reached 22.0 $\mu$mol $CO_2$ m$^{-2}$ s$^{-1}$, which was 7.5 and 5.8 times higher than for RT and NT, respectively.

Suitable weather conditions can increase $CO_2$ emissions in late summer or early autumn, which was observed in the second half of September in the years 2019 and 2020

and in the first half of September in the atypical year 2021 (Figure 4c–e). In the year 2021, the weather conditions delayed the soil tillage and measurements of $CO_2$ emissions, which started on 25 August, later than in precedent years. Warmer weather in September 2021 (Figure 4e) increased $CO_2$ emissions mainly under conventional tillage. The year 2022, with its warm weather and frequent precipitation, led to the typical pattern in which the highest $CO_2$ emissions were found under CT and were lower under RT and NT. The temperature and precipitation conditions in September and the first half of October 2022 were relatively high (Figure 4f), which enabled higher $CO_2$ emissions in this period.

## 4. Discussion

$CO_2$ emissions in soils under different tillage practices were measured in the August–October period in the years 2017–2022. Soil tillage in summer carried out shortly after the harvest of the main crop is practiced mainly due to the necessity of soil preparation for early-sown winter crops (i.e., oilseed rape) or intercrops. Typical August mean daily temperatures in the Czech Republic often exceed the summer temperature of 25 °C and in the afternoon become tropical, with a Tmax that is $\geq$30.0 °C. These conditions in combination with soil tillage can significantly increase the $CO_2$ emissions from soil. It was shown after winter wheat harvest that they depend on the intensity of the soil tillage [9], the current weather conditions, the soil water content, and the temperature [15].

The increased temperatures in the Czech Republic in the years 2000–2019 in comparison to 1961–2000 in combination with the lack of precipitation (namely winters without snow) caused a decrease in the available soil water. This caused widespread shifts in agroclimatic zones, which due to warmer and drier conditions resulted in decreased production potential [28]. The intensive soil cultivation in the warm summer period caused higher oxygenation of soils leading to higher mineralization of organic matter, losses of soil carbon, and increased $CO_2$ emissions [11,13]. However, in the case of wheat-based systems, the soil organic carbon mineralization rates were lower in comparison with mustard [13].

In the years 2015, 2018, and 2019, drought affected almost the entire Czech Republic. The droughts in 2000, 2004, 2007, 2012, 2014, and 2017 had smaller extents but still had severe intensities [4], which in some regions resulted in substantial winter wheat yield decreases [29]. Similarly, Thaler et al. [30] found a yield decrease under rainout shelters and stated that soil moisture played a significant role in yields, with a predictive model based on the monthly soil moisture explaining up to 79% (winter rape) of the yield variance.

The moisture and temperature conditions also affected the $CO_2$ emissions from soils in our field experiment. The $CO_2$ emissions differed in each of the study years from 2017–2022, with the lowest $CO_2$ emissions being found in 2018. The year 2018 was characterized by the highest number of summer days (Tmax $\geq$ 25.0 °C) and was among four years with the highest number of tropical days (Tmax $\geq$ 30.0 °C) since the year 1990 [6]. Panagea et al. [31] stated that soil water retention was an important soil property related to soil structure, texture, and soil organic matter, which consequently has an indirect impact on higher matric potentials, soil structure, and aggregate composition. The heat waves and drought in our field experiment could have caused changes in the soil structure in the form of destruction of soil aggregates [8,32] creating dust during soil cultivation. This phenomenon could have affected the stability of soil aggregates, changed the composition of soil microbial communities, and further impacted the soil carbon cycling processes [33] in the hot and dry year of 2018. These conditions affected the $CO_2$ emissions from soils despite the fact that some rainfall (40 mm) was noted in the two weeks before the start of $CO_2$ measurements. However, rewetting following a longer drought should not be sufficient to increase $CO_2$ emissions [34]. In fact, a longer heat wave duration may also cause death in soil microbial communities, and their reconstruction is associated with a shift in physiological traits [35]. Relatively regular precipitation in August 2019 and high precipitation events at the beginning of September 2019 probably led to the highest measured $CO_2$ emissions under the CT (21.5 µmol $CO_2$ m$^{-2}$ s$^{-1}$ in comparison with RT and NT reaching only 3.2 and 4.3 µmol $CO_2$ m$^{-2}$ s$^{-1}$, respectively, and averages of 26.8. and 3.9 in 2019). On the other hand, higher



precipitation in the year 2021 decreased the mean temperature in August in comparison with the 1995–2014 temperature average. The warmer weather in September (Figure 1) caused higher $CO_2$ emissions in September than in August. Warmer weather in autumn can appear more often in ongoing climate change conditions. As a consequence, the $CO_2$ emissions can also remain higher in this yearly phase.

In our experiment, we found a mean decrease of 45% in $CO_2$ emissions for RT and 51% for NT compared to CT, with an annual variability of 5.6–70.8% for RT and 17–79.3% for NT. These results are in good agreement with Sosulski et al. [36] who revealed that reduced tillage on sandy soil resulted in a reduction in $CO_2$ emissions from the soil by 28.7–61.2% in normal and drought weather, respectively. Similarly, Chatskikh and Olesen [37] reported $CO_2$ emissions under spring barley decreasing with a lower intensity of soil tillage: compared with CT (40 kg C day$^{-1}$), the cumulated $CO_2$ emissions during the 91 days after tillage were 21 and 25% lower for the RT and the DD (direct drilling) treatments, respectively. The study period of 2017–2022 partly belongs to the prolonged drought episode of 2015–2019, which included several seasons of agricultural droughts in the Czech Republic [4]. The compounded effect of consecutive droughts in agriculture systems without any drought insurance has led to profound economic problems across the agricultural sector [4]. The increasing trends in temperature and significant changes in the occurrence of hot temperature waves [6] accentuate the necessity of changes in soil tillage towards less intensive practices.

The negative effects of intensive soil tillage after the winter wheat harvest can be mitigated by reducing the tillage intensity, which can increase the soil aggregates' weight diameter and the soil's organic matter content [38] and lead to decreased $CO_2$ emissions from soils. Our field trial showed that the type of tillage affected $CO_2$ emissions mainly shortly after the soil tillage that was carried out in the summer period (Figure 3).

The $CO_2$ emissions were on average higher under CT and decreased with lowering intensity of soil tillage in the following order: CT > RT > NT. Generally, the plowed soils have better access to oxygen mainly after soil cultivation, which supports the mineralization of soil organic matter and consequently leads to higher $CO_2$ emissions. Higher $CO_2$ emissions were usually measured under RT in comparison with NT. However, in 2017 and 2019 and to a lesser extent in 2018, some measurements of $CO_2$ emissions were higher under NT in comparison with RT. In these cases, appropriate conditions could cause a possible decomposition of the straw on the straw–soil boundary in NT, resulting in increased $CO_2$ emissions. In addition, crop sowing can result in partial soil aeration in NT, which can also increase $CO_2$ emissions. In fact, wheat straw decomposition depends on the manner of its incorporation or leaving on the soil surface, as well as on the current weather conditions [39].

The soil moisture was the lowest under the CT tillage practice and mostly the highest under NT. Wheat straw mulch under NT is able to prevent water evaporation from soils, to maintain higher soil moisture, and to prevent soil overheating [40], which in the end decrease soil mineralization and losses of soil organic matter. In our study, the lowest carbon losses from the soil through $CO_2$ emissions and best water retention were found under RT and mainly NT. Therefore, if it is not necessary to process the soil for sowing rapeseed or a catch crop, it is preferable to preserve the soil in the warm summer period and limit its heating and aeration.

Significant correlations were found between soil moisture and $CO_2$ emissions under CT and RT, whereas no significant relationships were found for NT. Maize [30] and wheat [41] straw mulch under the NT system prevent C from mineralizing, prevent water evaporation from soils, and prevent soil overheating [40]. On the other hand, Yao et al. [42] reported that soil under straw mulch had higher annual mean $CO_2$ emissions than soil under CT treatments; however, they concluded that considering the future changes in temperature, precipitation, and atmospheric $CO_2$ concentrations, no-tillage practices have better potential to mitigate soil $CO_2$ emissions.

The use of reduced and no-tillage practices in the summer period, the most critical period for $CO_2$ and water losses from soils, will contribute to the mitigation of $CO_2$ emissions required by the European Union. This research was carried out more than 20 years following the beginning of the application of various soil tillage practices. The results obtained in our field experiment contribute to the choice and application of appropriate soil management practices having the potential to mitigate $CO_2$ emissions and to build carbon stock in soils at the national and European levels [25]. The $CO_2$ emission factors given by the Commission Decision [24] should be adjusted for the critical summer tillage.

## 5. Conclusions

This research focused on the evaluation of the most vulnerable period in terms of $CO_2$ emissions from soils after summer soil tillage in a more-than-20-year field experiment. There is still a lack of data for this period, which is important from the point of view of $CO_2$ and soil water losses following summer soil tillage. The six-year study showed the variation in $CO_2$ emissions under different tillage systems carried out in the summer period and under different weather conditions:

- A lower intensity of soil tillage decreased the $CO_2$ emissions under the reduced and no-tillage practices in comparison with conventional tillage on average by 45 and 51%, respectively.
- The current weather conditions, mainly temperature and precipitation, played an important role in $CO_2$ emissions during the study period. The hot and dry weather in 2018 decreased overall $CO_2$ emissions, which even under conventional tillage did not exceed 6 $\mu$mol $CO_2$ m$^{-2}$ s$^{-1}$.
- The atypical weather in the year 2021, with a colder August by about 2 °C in comparison with the long-term average, led to delays in the soil operations, lower $CO_2$ emissions at the end of August, and an increase in emissions in the warmer September 2021.
- These conditions will place higher demands on agronomists and their decisions regarding soil tillage. Reducing the soil tillage and the remaining soil being covered with straw mulch retained more soil moisture than did conventional tillage and also reduced $CO_2$ emissions from soils.
- The obtained data will contribute to the choice and application of appropriate soil management practices having the potential to mitigate $CO_2$ emissions and to build carbon stock in soils at the national and European levels.
- The obtained results will serve also at the national and European levels to adjust the $CO_2$ emission factors, which are given generally without emphasis on the warm summer period, which is the most vulnerable to $CO_2$ emissions.

**Author Contributions:** Conceptualization, G.M. and P.R.; methodology, G.M.; software, G.M.; validation, P.R. and H.K.; formal analysis, G.M., H.K. and R.V.; investigation, G.M., H.K. and R.V.; resources, G.M., H.K. and R.V.; data curation, G.M. and H.K.; writing—original draft preparation, G.M.; writing—review and editing P.R. and H.K.; visualization, G.M.; supervision, P.R.; project administration, P.R.; funding acquisition, P.R. All authors have read and agreed to the published version of the manuscript.

**Funding:** This research was funded by the NAZV ČR Agency no: QK21020121 and the Ministry of Agriculture of the Czech Republic MZE-RO0423.

**Data Availability Statement:** The original datasets are not publicly available due to the disagreement of all the authors in terms of solving the ongoing project.

**Acknowledgments:** The authors thank Martin Káš and Dana Hejnová for the technical support.

**Conflicts of Interest:** The authors declare no conflict of interest.

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
