# Peer review of "CO2 Emissions from Soils under Different Tillage Practices and Weather Conditions"

_agronomy, doi:10.3390/agronomy13123084_

Round 1

Reviewer 1 Report

Comments and Suggestions for Authors

The topic is relevant, but this manuscript is very lacking in novelty. A large number of studies of this kind have been carried out in the European Union. A large number of publications have been published in high-profile journals. Therefore, in order to publish this manuscript, it is necessary to focus on highlighting the novelty of the work and identifying new research results.

Other observations are given below:

·       Line 17. What do the letters CRI mean? For the first time, it is necessary to indicate in full words in the text.

·       The aim of the work is not given in the abstract.

·       Line 22. Incorrectly CO2

·       Line 22. After 5.8  dot is not needed.

·       Line 34. The abbreviation IPCC should be presented in full words for the first time.

·       Line 57. What is meant by "The improved tillage practices..."?

·       At the end of the introduction, it is necessary to state the novelty of this work and present the aim of the work.

·       Line 109-110. What were the fertilizer rates? What other fertilizers were used?

·       What kind of spraying was carried out, what pesticides were used? Do they have an impact on CO2 emissions?

·       Line 113. What does shortly after tillage mean? Based on the results of various authors, it is known that the first minutes after tillage are the most important for CO2 emissions.

·       Subsection 2.3. How were the measurement sites chosen? Were the measurements taken at the same locations in the field all the time? What measuring rings were used?

·       Fig.1 and Fig.2 Why do you need the word "Year" on the left side at the bottom? What is the meaning?

·       Line 160. What soil functions were affected?

·       In the discussion section, comparisons of the results with those obtained by other authors were made, but it is not indicated whether the same type of plants were grown.

·       In the conclusions, it is necessary to indicate what new results were obtained, which have not been published by other authors.

Author Response

Manuscript: agronomy-2748746: CO2 emissions from soils under different tillage practices and weather conditions.

Authors: Gabriela Mühlbachová, Pavel Růžek, Helena Kusá and Radek Vavera

Authors thank to the Referee 1 for his valuable comments on our manuscript which helped us to improve the manuscript.  We thank also editorial board for the evaluation of the manuscript.

Response to Referee 1

The topic is relevant, but this manuscript is very lacking in novelty. A large number of studies of this kind have been carried out in the European Union. A large number of publications have been published in high-profile journals. Therefore, in order to publish this manuscript, it is necessary to focus on highlighting the novelty of the work and identifying new research results.

Answer: The novelty of the research:

The research was carried out after more than 20 years since the beginning of the application of various soil tillage practices and continued for six years.  This length is unique in the Czech Republic. The summer period is the most vulnerable for CO2 emissions from soils under tillage. This research evaluated CO2 emissions in a six years period under weather conditions of a given year in order to cover the weather and CO2 emissions variability  There are still not enough data on CO2 for tillage in the critical summer period. During the summer, temperatures are high and deeper soil cultivation necessary for crops needing early summer sowing results (I.E. oilseed rape) in high CO2 emissions and water losses from the soil. Our research responds to the request of our Ministry of Agriculture and the European Commission to specify CO2 emissions in the summer period, for which sufficient data on CO2 emissions after long-term different soil tillage are not still available. The achieved results will thus serve as a basis for an eventual adjusting of CO2 emission factors for soil tillage practices in summer period given in EU Commission Decision from the year 2010. We added more information in the relevant parts of the manuscript (Abstract, Introduction, Dscussion and Conclusions).

Other observations are given below:

  • Line 17. What do the letters CRI mean? For the first time, it is necessary to indicate in full words in the text.

Answer: Line 23: CRI means „Crop Research Institute“. The abbreviation was substituted by full name of out institute.

  • The aim of the work is not given in the abstract.

Answer: Lines 10-14: The aim of the work was added in the abstract

  • Line 22. Incorrectly CO2

Answer: Line 29: The CO2 was corrected

  • Line 22. After 5.8  dot is not needed.

Answer: Line ……… The dot was eliminated.

  • Line 34. The abbreviation IPCC should be presented in full words for the first time.

Answer:  Line 41: The full name „Intergovernmental Panel on Climate Change” (IPCC) was added.

  • Line 57. What is meant by "The improved tillage practices..."?

Answer: Line 29: The word „improved“ was substituted to „conservative“

  • At the end of the introduction, it is necessary to state the novelty of this work and present the aim of the work.

Answer: Line 104-109: The aims of the research were improved.

  • Line 109-110. What were the fertilizer rates? What other fertilizers were used?

Answer: Line 136-138: Fertilizer doses were as follows: phosphorus as diammonium phosphate – 26,4 kg P2O5 ha-1, potassium as potassium chloride – 40 kg K2O ha-1.

  • What kind of spraying was carried out, what pesticides were used? Do they have an impact on CO2emissions?

Answer:  Lines 140-143: The pesticides were sprayed on the whole experimental field during the winter wheat growing, the last application was carried out at BBCH 37 at least 2 – 2,5 month before the harvest. No pesticides or fungicides were applied after the soil tillage. Short description was added in Materials and Methods.

  • Line 113. What does shortly after tillage mean? Based on the results of various authors, it is known that the first minutes after tillage are the most important for CO2emissions.

Answer: Line 146: The expression “shortly after tillage" was removed. The start of measurements one or two days after tillage is given after the dates of soil tillage.

We agree that the CO2 emissions can increase immediately after soil tillage, however from the technology point of view, there is necessary to wait the end of soil operations in the field and leaving the machines. In addition, there is necessary to install the rings which installation can during the first minutes and hours unpredictably affect the CO2 emissions. There is necessary to stabilize and equilibrate the soil conditions after the installing the rings into a soil to be sure that the CO2 emissions were result of the given tillage technology and not the disturbance of installing of rings.   

  • Subsection 2.3. How were the measurement sites chosen? Were the measurements taken at the same locations in the field all the time? What measuring rings were used?

Answer:  Lines 150-155: The following information was added in Subsection 2.3.:  The measuring sites were chosen during the winter wheat vegetation when the evenness of the wheat growth was evaluated in order to avoid the damaged places. The rings were placed also according to the movement of machinery so that to avoid the paths overlapping and wheel tracks. The same measuring locations were taken in the field during the whole measuring period after the winter wheat harvest. The original LI-COR rings were used.

  • Fig.1 and Fig.2 Why do you need the word "Year" on the left side at the bottom? What is the meaning?

 Answer: Fig.1 and Fig.2  - The word “Year” was changed to Month which was given under the X axis  – it is description of the X axis. We are really sorry for the wrong expression.

  • Line 160. What soil functions were affected?

Answer: Line 200: The sentence was improved  …the microbial activities and soil mineralization processes in soils.

      In the discussion section, comparisons of the results with those obtained by other authors were made, but it is not indicated whether the same type of plants were grown.

Answer: (Lines 336, 345-346, 384, 394, 408, 412, 421(: The type of crops was added in the discussion section where relevant

  • In the conclusions, it is necessary to indicate what new results were obtained, which have not been published by other authors.

Answer: Lines 438-443, 460-465): The section Conclusions was improved. Our field experiment is unique as the research was focused on tillage carried out in summer period and there are compared three soil tillage practices in one field.  The early soil tillage in summer is the most risky in terms of the level of CO2 emissions from soils. We found a variability in CO2 emissions not only among soil technologies, but also within the study years which differed in weather conditions and consequently affected the CO2 emissions. Our research requested also our Ministry of Agriculture as a demand from European Commission.

Reviewer 2 Report

Comments and Suggestions for Authors

This study mainly investigated the effects of soil tillage practices on CO2 emissions in the six-year period in a long-term field experiment, established in the 1995, with different soil tillage practices; i.e., conventional tillage (CT; ploughing to 20–22 cm), reduced tillage (RT;  12 chiselling to 10 cm), no-tillage (NT; without tillage). The crop rotation is winter wheat-oilseed rape winter wheat-pea. After carefully reading, the following comments should be addressed to improve the manuscript:

(1)  Abstract: The main contributions of this study should be clarified in this section, please supplement them.

(2)  Introduction: The objectives or aims of the study and should be more clearly stated in this section, please explain them further.

(3)  Lines 50-51: ……increase lead to higher mineralization ……should be “……increase leads to higher mineralization ……”, please revise it.

(4)  Line 105: “field block in strips wide 12 m and 95 m long.” can be “field block in strips with width of 12 m and length of 95 m.”. Please revise it.

(5)  Section 2.2: The specific tool parameters in terms of both structure (e.g., tool width and rake angle) and working conditions (e.g., working speed) should be provided for both CT and RT practices, as they can significantly affect the soil disturbance and subsequent effects.

(6)  Tables 2-3: The meaning of the number of “*” should be clearly described at the end of each table to avoid confusion.

(7)  Fig.4c: Why was there a bar value at the lateral bar of 12.9 (i.e., 2,295), please check it.

(8)  Figs. 3-4: The title of horizontal axis should be provided in the figures, please revise them.

(9)  Conclusions: Please simplify this section by listing several points concluded from the results and discussion.

(10)                 The language should be improved.

Comments on the Quality of English Language

Moderate editing of English language is required.

Author Response

Manuscript: agronomy-2748746: CO2 emissions from soils under different tillage practices and weather conditions.

Authors: Gabriela Mühlbachová, Pavel Růžek, Helena Kusá and Radek Vavera

Authors thank to the referee 2 for his valuable comments on our manuscript which helped us to improve the manuscript.  We thank also editorial board for the evaluation of the manuscript and Ms. Holly Luan for managing the manuscript.

Response to Referee 2:

Comments and Suggestions for Authors

This study mainly investigated the effects of soil tillage practices on CO2 emissions in the six-year period in a long-term field experiment, established in the 1995, with different soil tillage practices; i.e., conventional tillage (CT; ploughing to 20–22 cm), reduced tillage (RT;  12 chiselling to 10 cm), no-tillage (NT; without tillage). The crop rotation is winter wheat-oilseed rape winter wheat-pea. After carefully reading, the following comments should be addressed to improve the manuscript:

Authors thank the Referee 2 for his valuable comments which enabled us to improve the manuscript. Our answers are given in following points:

  • Abstract: The main contributions of this study should be clarified in this section, please supplement them.

Answer: Lines 10-15: The aim and the main contributions were clarified in the abstract.

  • Introduction: The objectives or aims of the study and should be more clearly stated in this section, please explain them further.

Answer: Lines 104-109: Objectives of the study were improved at the end of the section Introduction.

  • Lines 50-51: “……increase lead to higher mineralization ……” should be “……increase leads to higher mineralization ……”, please revise it.

Answer: Line 59: “lead” was revised to “leads” – the expression was revised according the suggestion.

  • Line 105: “field block in strips wide 12 m and 95 m long.” can be “field block in strips with width of 12 m and length of 95 m.”. Please revise it.

Answer: Line 128:  field block in strips with width of 12 m and length of 95 m was revised according the suggestion

(5)  Section 2.2: The specific tool parameters in terms of both structure (e.g., tool width and rake angle) and working conditions (e.g., working speed) should be provided for both CT and RT practices, as they can significantly affect the soil disturbance and subsequent effects.

Answer: Lines: 128-130: The short characteristics of working machines was given in the  Subsection 2.2.

A reversible plough Kvernland ES 95 (Germany) with four ploughshares and clod crusher was used for mouldboard ploughing (CT). The working speed was 6 km h-1. The two-arm sweep cultivator Lemken Smaragd 7 (Germany) with five wing tines and cage roller was used for chiselling (RT). The working speed was 8 km h-1.

  • Tables 2-3: The meaning of the number of “*” should be clearly described at the end of each table to avoid confusion.

Answer: the  sense of “*” was added at the end of both Tables 2 and 3 - * = p < 0.05; ** = p < 0.01; *** = p < 0.001 

  • 4c: Why was there a bar value at the lateral bar of 12.9 (i.e., 2,295), please check it.

Answer: Fig. 4 c – the value was changed to the “abcde” expressing the statistical differences among tillage practices.

  • 3-4: The title of horizontal axis should be provided in the figures, please revise them.

Answer: The titles of horizontal axis were provided in the all figures 3 and 4. 

  • Conclusions: Please simplify this section by listing several points concluded from the results and discussion.

Answer: The main results in the section Conclusions were given in points in order to be more clear.

  • The language should be improved.

Answer:  The moderate English editing was performed.

Round 2

Reviewer 1 Report

Comments and Suggestions for Authors

The authors have done a good job of revising the manuscript based on the comments provided. In my opinion, the manuscript can be accepted for publication. Finally, more attention should be paid to the text. Spacing between rows is different, e.g. to line 71 and from line 72. The dashes between the numbers are sometimes short (line 139), long elsewhere (line 143), as well as in figures 1 and 2 between 1995 and 2014.

Reviewer 2 Report

Comments and Suggestions for Authors

All comments and suggestions have been addressed. Currently, the manuscript has been improved and can be considered to be accepted for publication.